# Aberrant FGF signaling promotes granule neuron precursor expansion in SHH subgroup infantile medulloblastoma

**Odessa R Yabut\*, Jessica Arela, Hector G Gomez, Jesse Garcia Castillo, Thomas Ngo, Samuel J Pleasure\***

Department of Neurology, Weill Institute for Neuroscience, University of California San Francisco, San Francisco, United States

## eLife Assessment

This study provides **valuable** new insight into the role of Fgf signaling in SUFU mutation-linked cerebellar tumors and indicates novel therapeutic interventions via inhibition of Fgf signaling. The potential impact of this work is therefore very high, and it is supported by **solid** evidence. However, due to current limitations in the full identification of the cell types secreting FGF5, and issues with robustness of evaluation of genetically engineered animals, the validation of some interpretations awaits future experiments.

**\*For correspondence:**
oryabut@gmail.com (ORY);
Samuel.Pleasure@ucsf.edu (SJP)

**Abstract** Mutations in Sonic Hedgehog (SHH) signaling pathway genes, for example, *Suppressor of Fused* (SUFU), drive granule neuron precursors (GNP) to form medulloblastomas (MB[SHH]). However, how different molecular lesions in the Shh pathway drive transformation is frequently unclear, and *SUFU* mutations in the cerebellum seem distinct. In this study, we show that fibroblast growth factor 5 (FGF5) signaling is integral for many infantile MB[SHH] cases and that *FGF5* expression is uniquely upregulated in infantile MB[SHH] tumors. Similarly, mice lacking SUFU (Sufu-cKO) ectopically express *Fgf5* specifically along the secondary fissure where GNPs harbor preneoplastic lesions and show that FGFR signaling is also ectopically activated in this region. Treatment with an FGFR antagonist rescues the severe GNP hyperplasia and restores cerebellar architecture. Thus, direct inhibition of FGF signaling may be a promising and novel therapeutic candidate for infantile MB[SHH].

## Introduction

Medulloblastoma (MB) is the most common malignant brain tumor in children, with half of the cases diagnosed before the age of 5 (*Ward et al., 2014*; *Ostrom et al., 2016*). Mutations in *Suppressor of Fused* (SUFU) comprise approximately 30% of tumors in infants with Sonic Hedgehog-driven MB (MB[SHH]). Infantile MB[SHH], including those associated with SUFU mutations (MB[SHH-SUFU]), has a worse prognosis and higher rates of local recurrence than other MB[SHH] subtypes (*Kool et al., 2014*; *Schwalbe et al., 2017*; *Guerrini-Rousseau et al., 2018*). Unfortunately, available SHH-targeted treatments for MB[SHH] act specifically on proteins upstream of SUFU and are therefore ineffective for MB[SHH-SUFU] patients (*Kool et al., 2014*). The poor prognosis, early occurrence, and lack of targeted therapy for MB[SHH-SUFU] patients make a detailed understanding of the drivers of oncogenesis in this group of great importance.

SUFU acts as an intracellular modulator of SHH signaling (*Matise and Wang, 2011*). Briefly, the Shh signaling pathway is initiated after the binding of extracellular Shh ligands to the transmembrane receptor Patched 1 (PTCH1). This relieves PTCH1 inhibition of the transmembrane protein,

Smoothened (SMO), and enables the initiation of a cascade of intracellular events promoting the activator function of the transcription factors, GLI1, GLI2, or GLI3. Sufu modulates Shh signaling by ensuring the stability of Gli transcription factors or by promoting the formation of the repressor forms of GLI2 (GLI2R) or GLI3 (GLI3R) (*Chen et al., 2009*; *Wang et al., 2010*; *Lin et al., 2014*). Thus, depending on the developmental context, loss of *Sufu* can lead to activation or repression of Shh signaling. In the developing cerebellum, SUFU dysfunction is associated with the abnormal development of granule neuron precursors (GNP), which account for MB[SHH] (*Kim et al., 2011*; *Kim et al., 2018*; *Vanner et al., 2014*; *Kong et al., 2019*; *Vladoiu et al., 2019*; *Yin et al., 2019*; *Jiwani et al., 2020*). GNPs populate the external granule layer (EGL) along the cerebellar perimeter, where local SHH signals trigger GNP proliferation and differentiation at neonatal stages. However, other mitogenic pathways can also influence GNP behavior (*Leto et al., 2016*), yet little is known about how SUFU interacts with these pathways. Understanding how *Sufu* loss of function (LOF) affects the activity of concurrent local signaling pathways in granule neuron development may be key to developing potent targets for anti-tumor therapy.

In this study, we examined the regulation of FGF signaling in MB[SHH] and identified upregulation of *FGF5* expression in tumors of infantile MB[SHH] patients. Similarly, we show ectopic expression of *Fgf5* in the neonatal cerebellum of mice lacking SUFU, which correlates with the activation of FGF signaling in surrounding EGL-localized cells where GNPs accumulate. Strikingly, acute pharmacological inhibition of FGF signaling results in near-complete rescue of these defects, including restoration of cerebellar

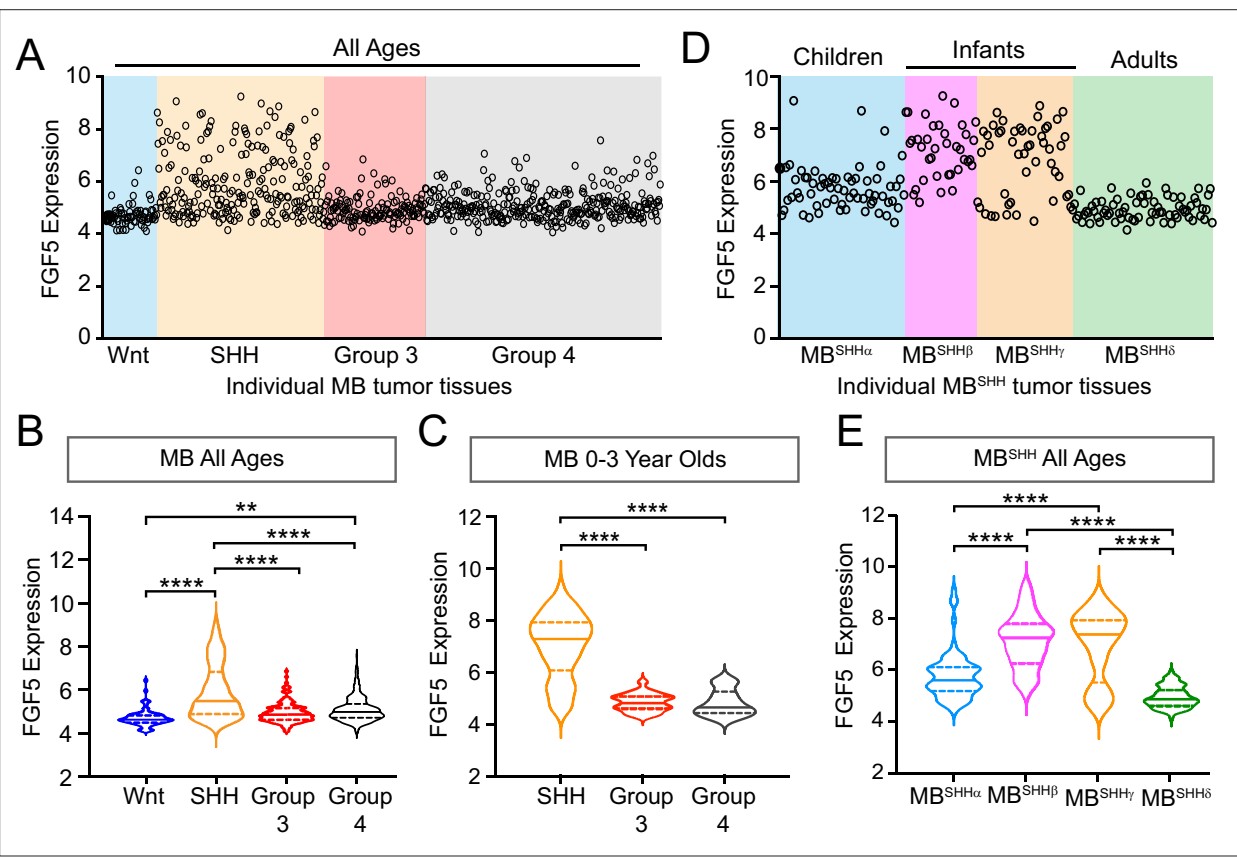

**Figure 1.** *FGF5* expression is upregulated in MB[SHH] tumors from infant patients. (**A**) Levels of *FGF5* expression in human medulloblastoma (MB) tumors of all ages from GEO expression dataset #GSE85217 (*Cavalli et al., 2017*). (**B, C**) Statistical analysis of FGF5 expression levels associated with MB tumor subtypes from patients across all ages (**B**) and 0–3 years old MB patients (**C**). **p<0.01, ****p<0.0001. (**D, E**) The graph represents *FGF5* expression levels in human MB[SHH] tumors of all ages from GEO expression dataset #GSE85217 (**D**) and corresponding plots (**E**) showing statistically higher *FGF5* expression in tumors from infants with MB[SHH] compared to tumors from children or adults with MB[SHH]. ****p<0.0001.

The online version of this article includes the following source data for figure 1:

**Source data 1.** Raw data for counts.

histo-organization. Thus, our findings identify *FGF5* as a potential biomarker for a subset of patients with infantile MB[SHH] who may be responsive to FGFR-targeting therapies.

## Results

### FGF5 is specifically upregulated in SHH-driven infantile MB

We previously reported that *Sufu* LOF in neocortical progenitors results in FGF signaling activation to influence the specification and differentiation of neocortical excitatory neurons (*Yabut et al., 2020*). Thus, we sought to determine if key FGF signaling pathway genes are differentially expressed in MB patient tumors. We performed a comparative analysis of the expression dataset from 763 MB patient samples comprised of tumors resected from molecularly distinct MB subgroups: Wingless (Wnt subgroup; MB[WNT]), MB[SHH], Group 3 (MB[Group3]), and Group 4 (MB[Group4]) (*Cavalli et al., 2017*). Strikingly, our analyses show that *FGF5* expression is higher in tumors, specifically from MB[SHH] patients, compared to other MB subgroups, with approximately 25% of MB[SHH] tumors exhibiting a twofold increase (*Figure 1A and B*). We also find that *FGF5* is uniquely upregulated in MB[SHH] tumors from patients within the 0–3-year-old age group, but not patients within the same age group in other MB subtypes (*Figure 1C*). Further examination across all MB[SHH] tumors stratified which subgroups express the highest levels of *FGF5* expression. Infantile tumors, largely belonging to the SHHβ and SHHγ subgroups (*Cavalli et al., 2017*), exhibit higher *FGF5* levels compared to tumors from children (SHHα) or adults (SHHδ) (*Figure 1D*). By all measures, the proportion of SHHβ and SHHγ tumors with relatively high levels of *FGF5* is significantly increased (~30%) compared to other MB[SHH] subgroups (*Figure 1E*). Taken together, these findings strongly suggest that FGF signaling is specifically disrupted in infantile-onset MB[SHH].

### Region-specific expansion of GNPs in the P0 Sufu-cKO cerebellum coincides with increased FGF5 expression

FGF signaling has been implicated in cerebellar development, particularly in granule neuron development (*Yaguchi et al., 2009*; *Yu et al., 2011*), leading us to wonder if and how aberrant FGF signaling may be contributing to the oncogenicity of GNPs. Since mutations in *SUFU* drive infantile MB[SHH], we generated the mutant mice (*hGFAP-Cre;Sufu[fl/fl]*, hereto referred to as Sufu-cKO), in which *Sufu* is conditionally deleted in GNPs (*Zhuo et al., 2001*) to examine FGF activity. Sufu-cKO mice exhibit profound defects in cerebellar development. At P0, a time point at which GNP proliferation and differentiation is ongoing, there is a visible increase in measured cerebellar size and expansion of Pax6-positive (Pax6+) GNPs in the EGL of Sufu-cKO cerebellum compared to controls (*Figure 2A–C*). Notably, the expansion of Pax6+ GNPs specifically localizes along the secondary fissure (EGL region B, arrow in *Figure 2A*) compared to other EGL areas (EGL regions A and C) in the P0 Sufu-cKO cerebellum (*Figure 2D*). We then proceeded to examine whether these defects correlated with abnormal FGF5 expression. We collected P0 littermates for these analyses and found that the foliation defects are particularly severe in the cerebelli of mice from this P0 Sufu-cKO litter compared to littermate controls (as determined by DAPI labeling in *Figure 2E*). In situ hybridization (ISH) using *Fgf5*-specific riboprobes show high expression of *Fgf5* (Fgf5[high]) immediately adjacent to the presumptive secondary fissure in the P0 control cerebellum (*Figure 2E*). Strikingly, in the P0 Sufu-cKO cerebellum, there is an expansion of Fgf5[high] expression regions (*Figure 2E*), coinciding with areas near the secondary fissure where GNP expansion is most severe (*Figure 2A and D*). Further, while *Fgf5*-expressing (FGF5+) cells are largely excluded from the EGL of the control cerebellum, a substantial number of *Fgf5*+ cells are ectopically localized in the EGL region B of the Sufu-cKO cerebellum (*Figure 2E and F*). *Fgf5* mRNA molecules are visibly higher in *Fgf5*-expressing cells in the Sufu-cKO EGL, while *Fgf5* expression is largely absent in the deeper EGL regions (outlined cells within boxed regions in *Figure 2F*). These findings implicate FGF5 as a potential instigator of region-specific defects in GNP differentiation present in the P0 Sufu-cKO cerebellum.

### FGF signaling drives GNP proliferation in the P0 Sufu-cKO cerebellum

FGF5 is a ligand for fibroblast growth factor receptors 1 (FGFR1) and 2 (FGFR2), both of which are expressed in the developing cerebellum, particularly in IGL regions where *Fgf5*-expressing cells localize (*Clements et al., 1993*; *Ornitz et al., 1996*). The binding of FGF5 to these receptors triggers the

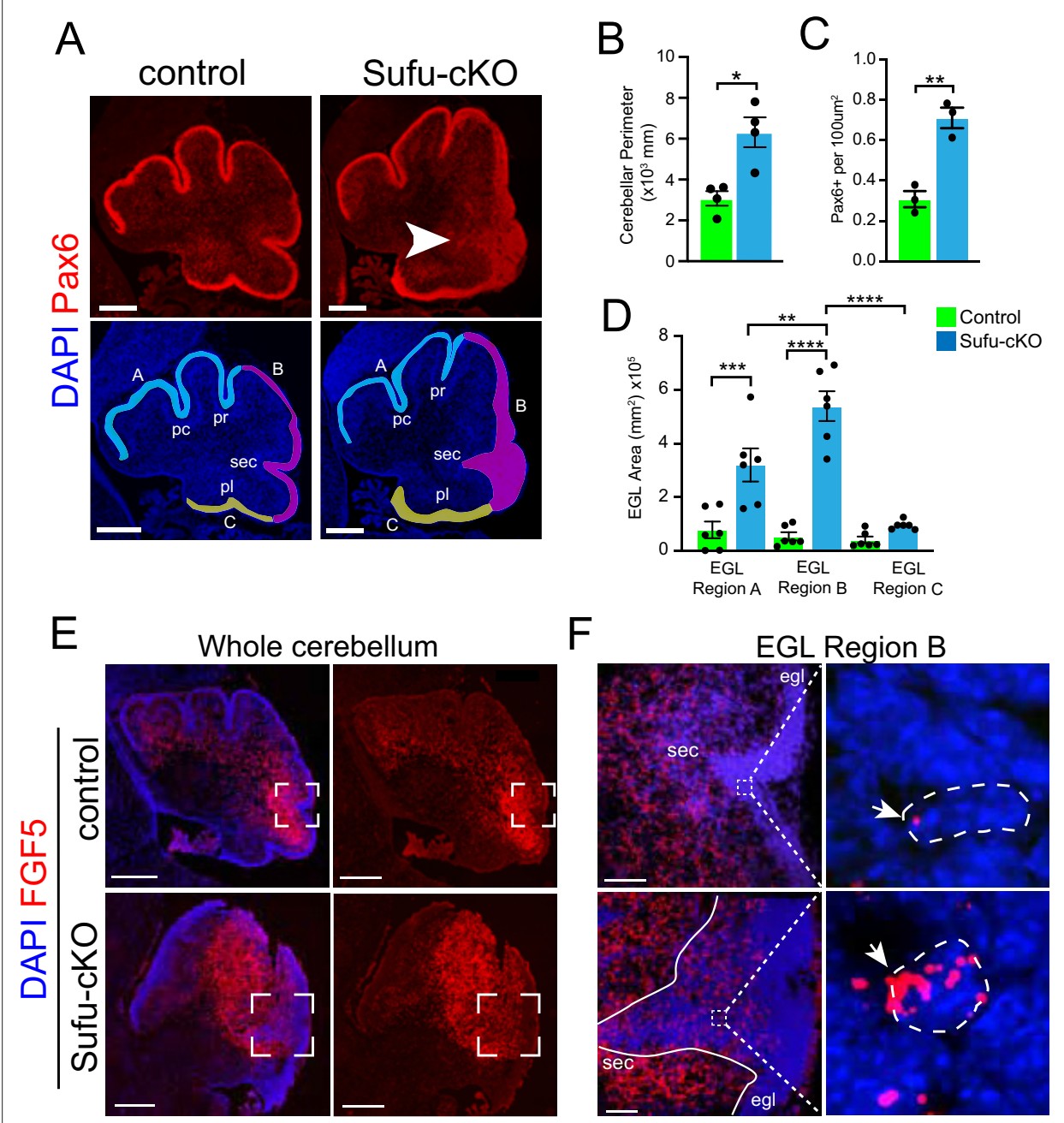

**Figure 2.** Increased *Fgf5* expression coincides with region-specific expansion of granule neuron precursors (GNPs) in the P0 Sufu-cKO cerebellum. (**A**) Pax6 (red) and DAPI (blue) immunofluorescence staining of the P0 Sufu-cKO and control cerebelli. Arrow points to severely expanded EGL region B in the P0 Sufu-cKO cerebellum. EGL regions are designated in DAPI-labeled sections as A (light blue), B (magenta), and C (yellow). Each region encompasses specific fissures: the preculminate (pc) and primary (pr) fissures for region A, the secondary (sec) fissure for region B, and the posterolateral (pl) fissure for region C. Scale bars: Scale bars = 250 µm. (**B–D**) Quantification and comparison of the cerebellar perimeter (**B**), total area occupied by densely packed Pax6+ cells (**C**), and size of specific EGL regions (**D**) between P0 Sufu-cKO and control cerebelli. (**E**) Fluorescent in situ hybridization using RNAScope probes against *Fgf5* mRNA (red) shows the expansion of *Fgf5* expression in the P0 Sufu-cKO cerebellum compared to controls. Sections are counterstained with DAPI to distinguish structures. Boxed areas are magnified in (**F**). Scale bars = 500 µm. (**F**) *Fgf5* is ectopically expressed in cells within the EGL of the P0 Sufu-cKO cerebellum. Boxed areas within the EGL show DAPI-labeled cells expressing visibly high levels of *Fgf5*, identified as punctate labeling (arrowheads), in the EGL of P0 Sufu-cKO cerebellum compared to controls. Scale bars = 50 µm. *p<0.05, **p<0.01, ***p<0.001, ****p<0.0001.

The online version of this article includes the following source data for figure 2:

**Source data 1.** Raw data for counts.

activation of multiple intracellular signaling pathways, including the mitogen-activated protein kinase (MAPK) pathway, to control cellular activities driving NSC progression (*Figure 3A*; *Ornitz and Itoh, 2015*). Immunostaining with antibodies to detect MAPK pathway activation reveals increased MAPK activity in the EGL of the P0 Sufu-cKO cerebellum. Regional distribution of cells labeled with phospho-Erk1/2 (pErk1/2+), a marker for activated MAPK signaling, shows that the abnormally expanded EGL of the secondary fissure in the P0 Sufu-cKO cerebellum increase in these cells (*Figure 3B*). We quantified the number of pERK1/2+ cells in proliferative regions defined by dense Ki-67-labeled cells and found a significant increase in pERK1/2+ cells compared to controls (*Figure 3C*). Many pERK1/2+ cells are also proliferative as indicated by Ki-67 labeling (boxed areas, *Figure 3B*) and the numbers of these dual labeled cells were significantly higher in the Sufu-cKO cerebellum compared to controls (*Figure 3D*). These findings indicate that the increase in *Fgf5* expression correlates with the activation of FGF signaling in GNPs and demonstrates a likely role in regulating the abnormal proliferation and pre-neoplastic lesion in mutant mice.

Given the role of SUFU in regulating GLI transcription factors to modulate SHH signaling activity (*Kim et al., 2018*; *Yin et al., 2019*), we examined whether the activation of FGF signaling occurred concurrently with SHH signaling activation. Surprisingly, GLI protein levels in the P0 control and Sufu-cKO cerebellum show a marked reduction of GLI1, GLI2, GLI3, and PTCH1 levels (*Figure 3—figure supplement 1*); this is inconsistent with elevated SHH signaling we anticipated in the absence of SUFU. To directly examine this in specific EGL regions, we compared the cerebellum of P0 control and Sufu-cKO mice carrying the Shh reporter transgene *Gli1-LacZ* (*Bai et al., 2002*). In these mice, Shh signaling activity is absent or very low throughout the entire P0 Sufu-cKO cerebellum but is highly active in a region-specific manner in controls (*Figure 3—figure supplement 2*). Furthermore, while some LacZ+ cells are detectable in EGL regions A and C and adjacent ML and IGL, LacZ+ cells are completely absent in EGL region B and adjacent areas of the P0 Sufu-cKO cerebellum (*Figure 3—figure supplement 1B*). These findings indicate that the accumulation of GNPs does not rely on active Shh signaling, particularly in Region B, where there is a severe expansion of GNPs in the absence of SUFU.

## Blockade of FGF signaling dramatically rescues the Sufu-cKO phenotype

To determine if activated FGF signaling drives GNP defects in the Sufu-cKO cerebellum, we pharmacologically inhibited FGF signaling using the competitive FGFR1-3 antagonist AZD4547 (*Gudernova et al., 2016*). For this experiment, 1 µl of AZD4547 (5 mg/ml) was delivered via intraventricular (IV) injection for five consecutive days beginning at P0, and the cerebellum was analyzed 3 days after treatment at P7 (*Figure 3E*). Strikingly, AZD4547 treatment results in near complete rescue of the GNP phenotype by P7 in the Sufu-cKO cerebellum, displaying a cerebellar morphology indistinguishable from controls with normal foliation and cellular organization (*Figure 3F*). Indeed, in the cerebellum of AZD4547 -treated P7 Sufu-cKO mice, proliferating Ki-67+ cells largely exclusively localize in the EGL while NeuN+ cells densely pack the IGL and not the EGL (*Figure 3G*). Notably, NeuN expression appears in cells localized at the border of the EGL and ML, where Ki-67+ cells are absent, indicating that post-mitotic cells successfully began differentiation as observed in controls (boxed regions, *Figure 3G*). Thus, our findings confirm that inhibition of FGF signaling in proliferating GNPs of the Sufu-cKO cerebellum ensure normal progression of GNP differentiation.

## *Fgf5* expression is increased in the developing cerebellum of Sufu;Trp53-dKO mice

Our findings indicate a critical role for FGF signaling in driving GNP hyperplasia, making GNPs vulnerable to neoplastic lesions, resulting in tumorigenesis when SUFU is absent in the developing cerebellum. Indeed, we find that Pax6+ GNPs within the neonatal Sufu-cKO EGL display an increase in double-strand breaks, especially in region B (DSBs; *Figure 4A and B*), as detected by immunostaining for phosphorylated H2AX (γH2AX), an early marker for DSBs (*Mah et al., 2010*). Nevertheless, as previously reported, tumors do not readily form in the Sufu-cKO cerebellum (*Yin et al., 2019*), indicating either timely repair of DSBs or the induction of apoptosis in GNPs with significant genomic instability. Indeed, double-labeling with cleaved Caspase 3 (CC3) and γH2AX shows a significantly higher number of double-labeled cells in EGL regions A and B (*Figure 4—figure supplement 1*).

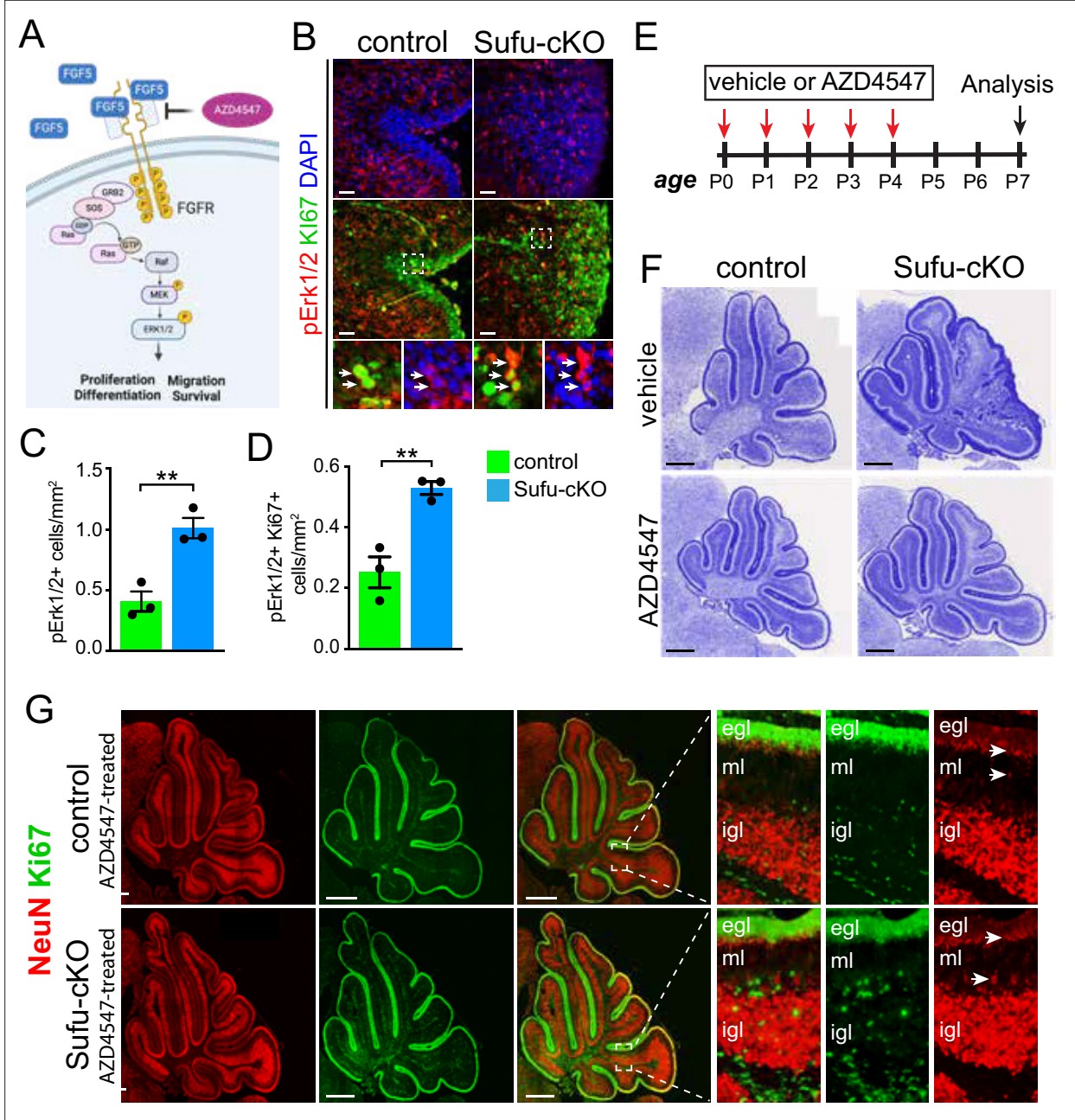

**Figure 3.** Ectopic activation of FGF signaling in the external granule layer (EGL) of P0 Sufu-cKO cerebellum. (**A**) Schematic diagram showing the activation of FGF signaling activity upon binding of FGF5 to extracellular domains of FGFR via the MAPK signal transduction pathway. Created with BioRender. (**B**) Double-immunofluorescence staining with Ki-67 (green) and phospho-Erk1/2 (pErk1/2; red), a marker of activated MAPK signaling in the P0 Sufu-cKO and control cerebelli. Boxed regions show pErk1/2+and Ki-67+ cells (arrowheads) in the control and Sufu-cKO EGL. Scale bars = 50 μm. (**C, D**) Quantification of pErk1/2+ cells (**C**) and double-labeled pErk1/2+and Ki-67+ cells (**D**) in the P0 Sufu-cKO and control EGL region B. **p<0.01. (**E**) Experimental design of rescue studies performed by intraventricular administration of FGFR1-3 pharmacological inhibitor, AZD4547, or vehicle controls. (**F**) Nissl staining of the P7 control and Sufu-cKO treated with either AZD4547 or vehicle, 2 days after treatment. Scale bars = 500 μm. (**G**) NeuN and Ki-67 double immunofluorescence staining of the P7 control and Sufu-cKO treated with AZD4547. Boxed regions show localization and organization of NeuN+ and Ki-67+ cells in distinct cerebellar layers. Arrows point to areas of the EGL and IGL where NeuN+ cells are beginning to be expressed.

The online version of this article includes the following source data and figure supplement(s) for figure 3:

**Source data 1.** Raw data for counts.

**Figure supplement 1.** Reduced SHH signaling activity in the P0 Sufu-cKO cerebellum.

**Figure supplement 2.** Lac Z staining.

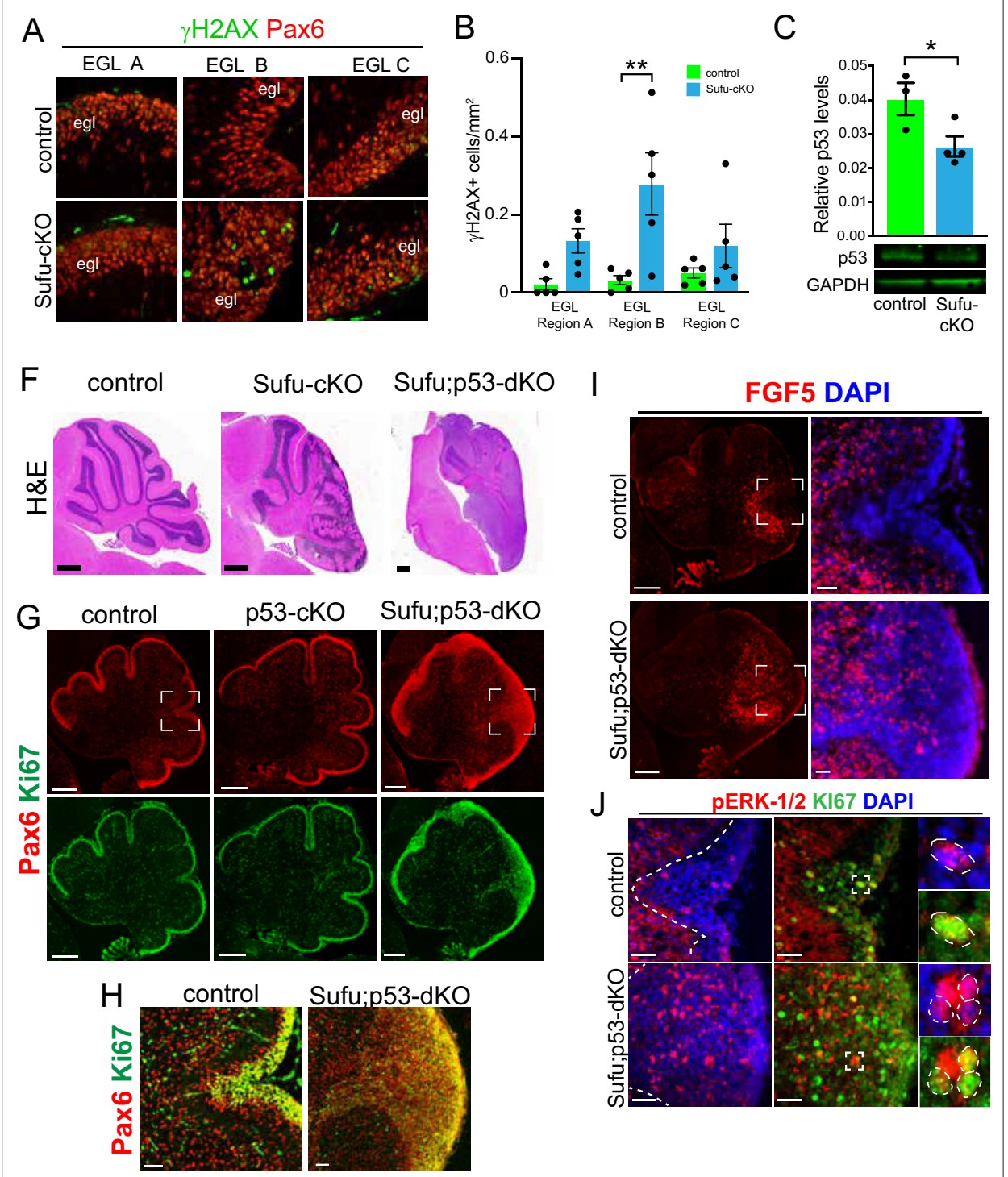

**Figure 4.** Evidence of pre-neoplastic lesions and high rates of cell death in Sufu-cKO granule neuron precursors. (**A**) Double-immunofluorescence staining with Pax6 (red) and γH2AX (green), a marker for double-strand DNA breaks in specific external granule layer (EGL) regions of the P0 Sufu-cKO and control cerebella. (**B**) Quantification of γH2AX+ cells in each cerebellar region of P0 control and Sufu-cKO mice. **p<0.01. (**C**) Western blot analysis of Trp53 protein levels in P0 control and Sufu-cKO cerebellar protein lysates. *p<0.05. (**F**) H&E staining of P60 control, Sufu-cKO, Sufu;Trp53-dKO cerebella. Scale bars = 500 μm. (**G, H**) Double-immunofluorescence staining against Pax6 (red) and Ki-67 (green) in the P0 control, Trp53-cKO, and Sufu;Trp53-dKO cerebellum (**G**). Boxed regions in (**G**) are magnified in (**H**), demonstrating the expansion of the EGL in the P0 Sufu;Trp53-dKO cerebellum compared to littermate controls. Scale bars = 200 μm (**A**) and 50 μm (**B**). (**I**) Fluorescent in situ hybridization using RNAScope probes against *Fgf5* mRNA (red) and DAPI labeling in the P0 Sufu;Trp53-dKO and control cerebellum. Boxed areas are enlarged to show ectopic localization of *Fgf5*+

*Figure 4 continued on next page*

*Figure 4 continued*
cells in the EGL of the Sufu-Trp53-dKO cerebellum, unlike in controls. Scale bars = 200 μm and 50 μm (boxed area). **(J)** Double-immunofluorescence staining with Ki-67 (green) and phospho-Erk1/2 (pErk1/2; red) in the P0 Sufu;Trp53-dKO and control cerebelli. Boxed regions show cells double-labeled with pErk1/2+and Ki-67+ cells in the control and Sufu;Trp53-dKO EGL region B. Scale bars = 25 μm.

The online version of this article includes the following source data and figure supplement(s) for figure 4:

**Source data 1.** Raw data for counts.

**Figure supplement 1.** Evidence of pre-neoplastic lesions and high rates of cell death in Sufu-cKO granule neuron precursors.

Among the downstream targets of DSB repair pathways is Trp53, which, when activated, mediates cell death to suppress tumor formation (*Gao et al., 2000*). In the P0 Sufu-cKO cerebellum, Trp53 protein is present, albeit significantly reduced (*Figure 4C*). The reduction in Trp53 may be driving an increase in DSBs, yet it is still sufficient to induce apoptotic pathways. Supporting this, conditional ablation of both *Trp53* and *Sufu* in GNPs (*hGFAP-Cre;Sufu^{fl/fl};Trp53^{fl/fl}* or Sufu;Trp53-dKO) results in the formation of massive tumors in the cerebellum within 2 months after birth (*Figure 4F*), indicating the failure to activate critical apoptotic pathways. However, tumors do not form in mice lacking *Trp53* (*Marino et al., 2000*) or *Sufu* alone (*Figure 4F*). These findings suggest that in the absence of SUFU, the failure of GNPs to transition into fully differentiated granule neurons compromises genomic stability and renders GNPs extremely vulnerable to tumor formation with a second molecular hit.

We sought to confirm that upregulated FGF signaling also occurs in the tumor-prone Sufu;Trp53-dKO. As expected, the neonatal cerebellum of Sufu-cKO displays EGL expansion due to excess proliferative (Ki-67+) and Pax6+ cells in the P0 Sufu;Trp53-dKO cerebellum but not in Trp53-cKO and control cerebelli (*Figure 4G*). Further expansion of Pax6+ GNPs in the P0 Sufu;Trp53-dKO is also most severe along the secondary fissure (EGL region B) compared to other EGL areas (EGL regions A and C) of the P0 Sufu;Trp53-dkO cerebellum (*Figure 4G and H*). As with our observations in the P0 Sufu-cKO cerebellum, ISH for *Fgf5* in the P0 Sufu;Trp53-dKO cerebellum shows ectopic *Fgf5* expression. Particularly, *Fgf5*+ cells are expanded anteriorly and detected specifically around the secondary fissure of the P0 Sufu;Trp53-dKO cerebellum (*Figure 4I*). There is also ectopic MAPK signaling activity in the P0 Sufu;Trp53-dKO cerebellum, with significantly higher numbers of pErk1/2+ cells within the EGL, many of which are proliferative as marked by co-labeling with Ki-67, within the expanded EGL of the secondary fissure (*Figure 4J*). These findings indicate that, as in the P0 Sufu-cKO cerebellum, ectopic *Fgf5* expression triggers FGF signaling in GNPs in the P0 Sufu;Trp53-dKO cerebellum and may facilitate oncogenic transformation and tumor growth of GNPs.

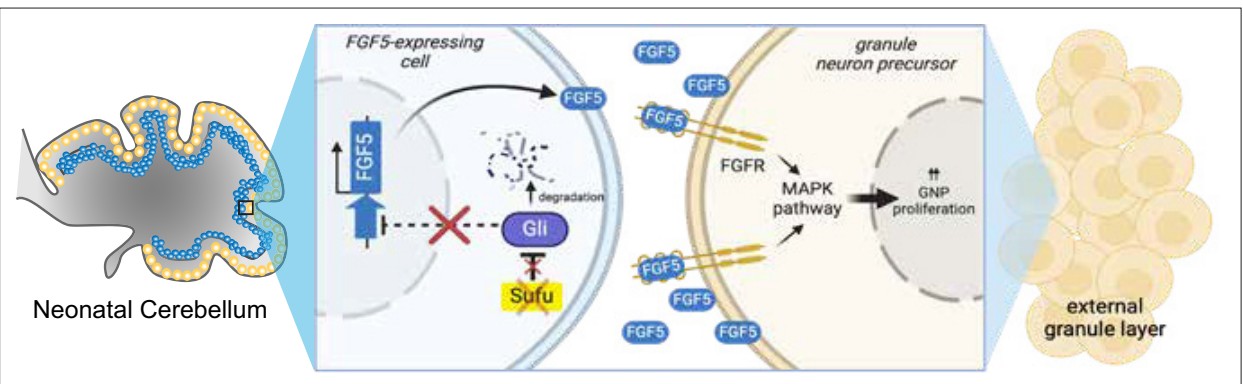

**Figure 5.** Loss of *Sufu* function drives excess proliferation of granule neuron precursors via FGF signaling activation. The schematic diagram models how *Sufu* loss of function (LOF) facilitates the expansion of granule neuron precursors (GNPs) (yellow cells) in the external granule layer (EGL) at the early stages of cerebellar development. We hypothesize that *Fgf5*-expressing cells (blue cells, yet to be identified) send FGF5 signals to GNP to proliferate and that ectopic expression of *Fgf5* in the absence of SUFU is responsible for the uncontrolled expansion of EGL localized GNPs. Created with BioRender..

## Discussion

Our studies identify a mechanism by which the combinatorial effects of oncogenic *SUFU* mutations and other concurrent developmental signaling pathways make GNPs vulnerable to oncogenic transformation leading to infantile MB[SHH]. Using mice lacking *Sufu* in GNPs, we find that ectopic *Fgf5* expression correlates with an increase in FGF signaling, particularly in areas where proliferating GNPs reside, resulting in GNP hyperplasia, preneoplastic lesions, and patterning defects. Inhibition of Fgf signaling through pharmacological blockade of FGFR1-3 prevents hyperplasia and associated cerebellar architectural abnormalities. Strongly supporting a role for FGF5, we also find elevated levels of *Fgf5* gene expression specifically in infantile MB[SHH] patients, but not in other MB subgroups. These findings indicate that FGF-targeting compounds may be a promising therapeutic option for infantile MB[SHH] patients, with elevated levels of FGF5 in tumor tissues acting as a potential biomarker.

Expansion of Pax6+ GNPs in the newborn Sufu-cKO cerebellum (*Figure 2*) occurs in the posterior/ventral regions of the cerebellar hemispheres where infantile MB tumors typically arise (*Tan et al., 2018*). Interestingly, these subregions have low levels of GLI1 reporter activity, PTCH1 expression, and SHH ligand expression (*Corrales et al., 2004*). Spatially distinct regulation of granule neuron development by SUFU may, therefore, rely on non-canonical SHH signaling beyond SMO or through yet undefined downstream interactions, resulting in control of FGF signaling activity. Supporting this, we find ectopic expression of *Fgf5* and deregulation of FGF signaling as marked by the excessive intracellular activation of MAPK signaling in the P0 Sufu-cKO cerebellum. Similarly, in previous studies, ectopic expression of FGF ligands, *Fgf8* and *Fgf15*, are associated with degradation of GLI3R because of *Sufu* deletion, resulting in regional patterning and precursor specification and differentiation defects (*Kim et al., 2011*; *Kim et al., 2018*; *Jiwani et al., 2020*; *Yabut et al., 2020*). Taken together, these findings indicate that SUFU acts at the intersection of SHH and FGF signaling to modulate GNP behavior (as summarized in our working model in *Figure 5*). We postulate that SUFU exerts this role via stabilization of GLI transcription factors since we observed a decrease in GLI1, GLI2, and GLI3 protein level in the newborn Sufu-cKO cerebellum. Supporting this, *Yin et al., 2019* found that reducing GLI2 levels rescued GNP hyperplasia and patterning defects in the Sufu-cKO neonatal cerebellum. Further studies are required to elucidate the involvement of GLI-dependent mechanisms in controlling *Fgf5* gene expression in the neonatal cerebellum.

In the wildtype cerebellum, *Fgf5*-expressing cells localize within the IGL surrounding the secondary fissure at P0 (*Figure 4B*; *Ozawa et al., 1996*; *Hattori et al., 1997*; *Yaguchi et al., 2009*) where GLI1 reporter activity (*Figure 4A*), and SHH and PTCH1 expression are lowest (*Corrales et al., 2004*). By P4, *Fgf5* expression is detected in cells within the IGL throughout the cerebellum (*Yaguchi et al., 2009*). However, by P14, a time point at which most granule neurons have differentiated, *Fgf5* is no longer detected in the IGL or other cerebellar regions per Allen Developing Mouse Brain Atlas (*Allen Institute for Brain Science, 2004*). The strict spatiotemporal expression of *Fgf5* strongly supports a stage-specific role for regulating GNP development, particularly along the secondary fissure, to ensure timely differentiation and maturation of granule neurons. Indeed, as we observed in the P0 Sufu-cKO cerebellum, deregulation of *Fgf5* expression correlates with the continued uncontrolled proliferation of GNPs that have failed to differentiate normally. Follow-up studies geared toward characterizing cell-specific expression of *Fgf5* and how FGF5 acts on GNP differentiation are crucial to solidifying the exact function of FGF5 in early neonatal cerebellar development.

Our findings are contrary to previous reports of the proliferation-suppressive roles of FGFs, particularly FGF2, in GNPs specifically carrying *Ptch1* mutations (*Fogarty et al., 2007*; *Emmenegger et al., 2013*). However, in contrast to the Ptch1-cKO cerebellum, the newborn Sufu-cKO cerebellum still expresses PTCH1 and exhibits reduced GLI1, GLI2, and GLI3 protein levels (*Figure 3—figure supplement 1*). These key molecular differences may activate unique signaling networks in Sufu-cKO GNPs. Additionally, 18 FGF ligands differ significantly in molecular features and binding specificities to distinct combinations of FGFR splice variants (*Ornitz and Itoh, 2015*). For example, unlike FGF2, which acts in an autocrine manner, FGF5 can exert both autocrine and paracrine functions, bind different combinations of FGFRs, and is dynamically expressed in distinct regions of the developing cerebellum. Thus, in-depth studies are needed to elucidate the exact mechanisms triggered by abnormally high levels of FGF5 in the developing cerebellum, particularly since *Fgf5* overexpression is known to drive cancer development and progression, including brain tumors (*Allerstorfer et al., 2008*). Importantly, examination of whether inhibition of FGF signaling by AZD4547 (or other FGFR

inhibitors) in MB tumor-bearing Sufu;Trp53-dKO mice is crucial to establish that FGF signaling is a druggable target for SUFU-associated infantile MB[SHH].

The high occurrence of *SUFU* mutations in infantile MB indicates the selective vulnerability of the developing cerebellum to the neoplastic effects of SUFU dysfunction. Notably, the timing of *SUFU* LOF is critical; conditional deletion of *Sufu* in neural stem cells before granule neuron specification (using the hGFAP-Cre line) results in GNP hyperplasia, whereas conditional deletion of *Sufu* after granule neuron specification (using the Atoh1-Cre line) does not lead to these defects (*Jiwani et al., 2020*). Thus, *SUFU*-associated infantile MB[SHH] is a likely consequence of defects stemming from the early stages of granule neuron lineage specification at embryonic stages. However, since hGFAP-Cre may also be expressed in cerebellar glial cells, we cannot yet eliminate the possibility that the defects are directly or indirectly a consequence of abnormal glial cell development and function. In-depth studies are required to interrogate the effects of SUFU LOF in cerebellar glial cell development and how this might indirectly affect GNP differentiation.

Unfortunately, tumors initiated at embryonic stages are typically undetectable until several months after birth, when tumorigenesis has significantly progressed. Thus, therapeutics for infantile MB must successfully curtail tumorigenic mechanisms at postnatal stages and minimally affect normal GNPs elsewhere in the cerebellum. Toward this goal, inhibiting localized FGF5 and FGF signaling activity may provide new paths toward the design of targeted treatments. Since we confirmed the occurrence of high *FGF5* levels in a subset of infantile MB[SHH] patients, measuring FGF5 may be a useful diagnostic biomarker for this patient population. This may predict the lack of efficacy of SHH-targeting compounds in curtailing tumor growth but could instead significantly impede cerebellar development. Importantly, detection of elevated FGF5 levels may identify patients who will be responsive to FGF-targeting treatments. Ultimately, we hope these studies facilitate the design of much-needed precision medicines to address the distinct oncogenic mechanisms specifically and effectively in infantile MB[SHH] patients while enabling normal progression of cerebellar development.

## Materials and methods
### Animals
Mice carrying the floxed *Sufu* allele (Sufu[fl]) were kindly provided by Dr. Chi-Chung Hui (University of Toronto) and were genotyped as described elsewhere (*Pospisilik et al., 2010*). *GFAP-cre*, a transgene driven by the human GFAP promoter (Stock #004600; *Schüller et al., 2008*), and Gli1-LacZ (Stock #008211) mice were obtained from Jackson Laboratories (Bar Harbor, ME). Mice designated as controls did not carry the *Cre* transgene and may have either one of the following genotypes: *Sufu*[fl/+] or *Sufu*[fl/fl]. All mouse lines were maintained in mixed strains, and the analysis included male and female pups from each age group, although sex differences were not included in data reporting. All animal protocols were in accordance with National Institutes of Health regulations and approved by the UCSF Institutional Animal Care and Use Committee (IACUC) (approvals AN200243-00B and AN204753-00).

### In vivo treatment with FGFR inhibitor
The FGFR1-3 inhibitor, AZD4547 (Selleck Chemicals, #S2801), was dissolved sequentially in 4% dimethyl sulfoxide (DMSO), 30% polyethylene glycol (PEG), 5% Tween-80, and water to make a 1 mM solution. 1 µl of 1 mM AZD4547, or vehicle only as control, was injected into the lateral ventricle (~1 mm from the cerebellar midline) of pups for 5 days from P0/P1 using a 2.5 µl syringe (Model 62 RN; Hamilton Scientific). Pups remained with the mother until perfusion at P7 for analysis.

### Immunohistochemistry and LacZ staining
Perfusion, dissection, immunofluorescence, and LacZ staining were conducted according to standard protocols as previously described (*Yabut et al., 2015*). Briefly, P0/P1 brain tissues were fixed after dissection by direct immersion in 4% paraformaldehyde (PFA) overnight. P7 and older postnatal brains were fixed by intracardial perfusion with 4% PFA followed by 2 hr post-fixation. All fixed brains were cryoprotected with a 15–30% sucrose gradient overnight before embedding in Optimal Cutting Temperature compound for cryosectioning. Cryostat sections were air dried and rinsed 3× in PBS plus 0.2%Triton before blocking for 1 hr in 10% normal lamb serum diluted in PBS with 0.2% Triton to prevent nonspecific binding. A heat-induced antigen retrieval protocol was performed on select

immunohistochemistry experiments using 10 µM citric acid at pH 6.0. Primary antibodies were diluted in 10% serum in PBS with 0.2% Triton containing 4'6-diamidino-2-phenylindole (DAPI); sections were incubated in primary antibody overnight at room temperature (RT). The following antibodies were used: rabbit anti-Pax6 (1:250 dilution; Cat# 901301, BioLegend); rabbit anti-NeuN (1:250 dilution; Cat# PA5-784-99, Invitrogen); mouse anti-Calretinin (1:250, Cat# AB5054, Millipore); rabbit anti-phospho-Erk1/2 (1:250 dilution; Cat# 4370, Cell Signaling); γH2AX (1:100 dilution; Cat# 05-636, Millipore); mouse anti-Ki-67 (1:100 dilution; Cat# 550809 BD Biosciences); and cleaved-Caspase 3 (1:250 dilution; Cat# 9661S, Cell Signaling). To detect primary antibodies, we used species-specific Alexa Fluor-conjugated secondary antibodies (1:500; Invitrogen) in 1× PBS-T for 1 hr at RT, washed with 1× PBS, and coverslipped with Fluoromount-G (SouthernBiotech).

## In situe hybridization

RNAScope ISH was conducted for *Fgf5* and *Ptch1*. RNAScope probes for Mm-*Fgf5* were designed commercially by the manufacturer (Cat# 417091, Advanced Cell Diagnostics, Inc). RNAScope Assay was performed using the RNAScope Multiplex Fluorescent Reagent Kit V2 according to the manufacturer's instructions with the following conditions. Slides of cryosectioned brain tissues were prepared by air-drying for 30 min at 60°C, then post-fixed in 4% PFA in 1× PBS for 15 min at 4°C. Tissues were dehydrated at RT in an ethanol gradient (50% ethanol, 70% ethanol, then 100% ethanol) for 5 min each, followed by a final 100% ethanol wash for 5 min. RNAScope Hydrogen Peroxide treatment of tissues occurred for 10 min. Target retrieval was performed by immersing slides in RNAScope 1× Target Retrieval Reagent for 5 min in a 99°C steamer. Tissue sections were subjected to RNAScope Protease Plus reagent treatment for 10–15 min at 40°C in HybEZ Oven (ACDBio). Tissues were hybridized with the probe mix and incubated for 2 hr at 40°C in the HybEZ Oven. AMP1, AMP2, and AMP3 sequential hybridization steps were performed as per manufacturer's instructions. Slides were incubated in RNAScope Multiplex FL v2 HRP-C1 for 15 min at 40°C. Signals were detected using the TSA Plus Fluorescein Kit (Cat# NEL741E001KT, PerkinElmer) and Opal Dye reagent (Opal 570, Cat# FP1488001KT or Opal 520, Cat# FP1487001KT, from Akoya Biosciences) after 30 min of incubation at 40°C.

## Western blot analysis

Western blot analyses were conducted according to standard protocols. Soluble extracts were loaded onto Criterion, 4–15% Tris-HCl 4 SDS-PAGE gels (Bio-Rad), separated at 120 V, and transferred to PVDF membrane at 30 V for 2 hr or overnight at 4°C. Membranes were blocked with 3% milk/1× TBS-T (Tris-buffered saline with 0.1% Tween 20) or 5% BSA/1× TBS-T for 1 hr at RT and incubated with primary antibodies diluted in blocking buffer overnight at 4°C, and secondary antibodies (1:5000 dilution; IR-Dye antibodies, LI-COR) for 1 hr at RT. Membranes were washed in 1× TBS-T and scanned using the Odyssey Infrared Imaging System (LI-COR). Primary antibodies were used as follows: rabbit anti-Gli1 (1:1000; Abcam); goat anti-Gli2 (1:1000; R&D Systems), rabbit anti-Gli3 (1:100; Santa Cruz), rabbit anti-Pax6 (1:1000 dilution; Cat# 901301, BioLegend); rabbit anti-NeuN (1:1000 dilution; Cat# PA5-784-99, Invitrogen); rabbit anti-GABA A Receptor α6 (1:1000 dilution; Cat# PA5-77403, GABRA6; Invitrogen), and α-Tubulin (1:5000 dilution; Cat# ab4074, Abcam). Quantification and analysis were conducted using the Odyssey Image Studio Software (LI-COR). Protein levels were normalized to GAPDH protein levels. Levels of NeuN and GABRA6 were quantified in correlation with Pax6 levels (NeuN/Pax6 or GABRA6/Pax6) to determine the proportion of Pax6+ cells expressing mature granule neuron markers.

## Image analysis and acquisition

Images were acquired using a Nikon E600 microscope equipped with a QCapture Pro camera (QImaging), Zeiss Axioscan Z.1 (Zeiss, Thornwood, NY) using the Zen 2 blue edition software (Zeiss), or the Nikon Ti inverted microscope with CSU-W1 large field of view confocal and Andor Zyla 4.2 sCMOS camera. All images were imported in tiff or jpeg format. Brightness, contrast, and background were adjusted equally for the entire image between controls and mutants using the 'Brightness/Contrast' and 'Levels' functions from the 'Image/Adjustment' options in Adobe Photoshop or NIH ImageJ without any further modification. NIH ImageJ was used to threshold background levels between controls and mutant tissues to quantify fluorescence labeling. To quantify cell density,

positively labeled cells within defined EGL regions, as defined in *Figure 2A*, were counted. Quantification of double-labeled cells was performed in 1–2 µm optical slices obtained by confocal microscopy. We relied on continuous DAPI nuclear staining to distinguish individual cells in each optical slice to determine the cellular colocalization of specific markers being analyzed (e.g., pERK and Ki-67). All measurements were performed from 2- to 3 20-µm-thick and histologically matched cerebellar sections of 3–6 independent mice per genotype analyzed. Individual points in the bar graphs represent the average cell number (quantified from 2 to 3 sections) from each mouse.

## Human MB gene expression

Expression values of FGF5 (ENSG00000138675) were assessed using Geo2R (*Barrett et al., 2013*) from published human MB subtype expression dataset accession no. GSE85217 (*Cavalli et al., 2017*). GEO2R is an interactive web tool that compares expression levels of genes of interest (GOI) between sample groups in the GEO series using original submitter-supplied processed data tables. We entered the GOI Ensembl ID and organized data sets according to age and MB subgroup or MB$^{SHH}$ subtype classifications. GEO2R results presented gene expression levels as a table ordered by FDR-adjusted (Benjamini and Hochberg) p-values, with significance level cutoff at 0.05, processed by GEO2R's built-in limma statistical test. The resulting data were subsequently exported into Prism (GraphPad). Scatter plots presenting FGF5 expression levels across all MB subgroups (*Figure 1A*) and MB$^{SHH}$ subtypes (*Figure 1D*). We performed additional statistical analyses to compare FGF5 expression levels between MB subgroups and MB$^{SHH}$ subtypes and graphed these data as violin plots (*Figure 1B, C and E*). For these analyses, we used one-way ANOVA with Holm–Sidak's multiple comparisons test, single pooled variance. p-value ≤0.05 was considered statistically significant. Graphs display the mean ± standard error of the mean (SEM). Sample sizes analyzed were MB$^{WNT}$ n = 70, MB$^{SHH}$ n = 224, MB$^{GR3}$ n = 143, MB$^{GR4}$ n = 326, MB$^{SHHα}$ n = 66, MB$^{SHHβ}$ n = 35, MB$^{SHHγ}$ n = 47, and MB$^{SHHδ}$ n = 76.

## Statistics

Prism 8.1 (GraphPad) was used for statistical analysis. Two sample experiments were analyzed using Student's *t*-test, and experiments with more than two parameters were analyzed by ANOVA. In one- or two-way ANOVA, when interactions were found, follow-up analyses were conducted for the relevant variables using Holm–Sidak's multiple comparisons test. All experiments were conducted at least in triplicate with sample sizes of n = 3–6 embryos/animals per genotype. p-value ≤0.05 was considered statistically significant. Graphs display the mean ± SEM.

## Acknowledgements

We thank Hirofumi Noguchi and other members of the Pleasure Lab for critical discussions, DeLaine Larsen and Kari Harrington at the University of California San Francisco Center for Advanced Light Microscopy for assistance with imaging, and William Krause for assistance with transcriptomics. Schematic diagrams were created with BioRender.com. This work was supported by the NIH R01MH077694 and R01NS118995 (SJP), R01MH077694-S1 (HG), NIH NIH/NCI K01CA201068 (ORY), and American Brain Tumor Association Grant #A131363 (ORY).

## Additional information

### Competing interests

Samuel J Pleasure: Reviewing editor, eLife. The other authors declare that no competing interests exist.

### Funding

| Funder | Grant reference number | Author |
| --- | --- | --- |
| National Institutes of Health | R01NS118995 | Samuel J Pleasure |

| Funder | Grant reference number | Author |
| --- | --- | --- |
| National Institutes of Health | K01CA201068 | Odessa R Yabut |
| National Institutes of Health | R01MH077694 | Samuel J Pleasure |
| National Institutes of Health | R01NS118995 | Samuel J Pleasure |
| National Institutes of Health | R01MH077694-S1 | Hector G Gomez |
| NIH/NCI | K01CA201068 | Odessa R Yabut |
| American Brain Tumor Association | A131363 | Odessa R Yabut |

The funders had no role in study design, data collection and interpretation, or the decision to submit the work for publication.

### Author contributions

Odessa R Yabut, Conceptualization, Formal analysis, Supervision, Funding acquisition, Investigation, Visualization, Methodology, Writing – original draft, Project administration, Writing – review and editing; Jessica Arela, Jesse Garcia Castillo, Investigation; Hector G Gomez, Investigation, Visualization; Thomas Ngo, Data curation, Investigation, Visualization; Samuel J Pleasure, Conceptualization, Resources, Supervision, Investigation, Project administration, Writing – review and editing

### Author ORCIDs

Hector G Gomez ⓘD https://orcid.org/0000-0002-8894-0093
Samuel J Pleasure ⓘD https://orcid.org/0000-0001-8599-1613

### Ethics

All animal protocols were in accordance with National Institutes of Health regulations and approved by the UCSF Institutional Animal Care and Use Committee (IACUC) (Approvals AN200243-00B and AN204753-00).

Reviewer #1 (Public review): https://doi.org/10.7554/eLife.100767.3.sa1
Reviewer #2 (Public review): https://doi.org/10.7554/eLife.100767.3.sa2
Reviewer #3 (Public review): https://doi.org/10.7554/eLife.100767.3.sa3
Author response https://doi.org/10.7554/eLife.100767.3.sa4

## Additional files

### Supplementary files
MDAR checklist

### Data availability

All data generated or analyzed in the study is included except the human sequences which were obtained from GEO databases.

The following previously published dataset was used:

| Author(s) | Year | Dataset title | Dataset URL | Database and Identifier |
| --- | --- | --- | --- | --- |
| Cavalli FM, Remke M, Taylor MD | 2017 | Expression data from primary medulloblastoma samples | https://www.ncbi.nlm.nih.gov/geo/query/acc.cgi?acc=GSE85217 | NCBI Gene Expression Omnibus, GSE85217 |

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
