## [Editor Report · eLife Assessment]

This study provides **valuable** new insight into the role of Fgf signaling in SUFU mutation-linked cerebellar tumors and indicates novel therapeutic interventions via inhibition of Fgf signaling. The potential impact of this work is therefore very high, and it is supported by **solid** evidence. However, due to current limitations in the full identification of the cell types secreting FGF5, and issues with robustness of evaluation of genetically engineered animals, the validation of some interpretations awaits future experiments.

---

## [Referee Report · Reviewer #1 (Public review)]

Summary:

SUFU modulates Sonic hedgehog (SHH) signaling and is frequently mutated in the B-subtype of SHH driven medulloblastoma. The B-subtype occurs mostly in infants, is often metastatic, and lacks specific treatment. Yabut et al. found Fgf5 was highly expressed in the B-subtype of SHH driven medulloblastoma by examining a published microarray expression dataset. They then investigated how Fgf5 functions in the cerebellum of mice that have embryonic Sufu loss of function. This loss was induced using the hGFAP-cre transgene, which is expressed multiple cell types in the developing cerebellum, including granule neuron precursors (GNPs) derived from the rhombic lip. By measuring the area of Pax6+ cells in the external granule cell layer (EGL) of Sufu-cKO mice at postnatal day 0, they find Pax6+ cells occupy a larger area in the posterior lobe adjacent to the secondary fissure, which is poorly defined. They show that Fgf5 RNA and phosphoErk1/2 immunostaining are also higher in the same disrupted region. Some of the phosphoErk1/2+ cells are proliferative in the Sufu-cKO. Western blot analysis of Gli proteins that modulate SHH signaling found reduced expression and absence of Gli1 activity in the region of cerebellar dysgenesis in Sufu-cKO mice. This suggests the GNP expansion in this region is independent of SHH signaling. Amazingly, intraventricular injection of the FGFR1-2 antagonist AZD4547 from P0-4 and examined histologoically at P7 found the treatment restored cytoarchitecture in the cerebella of Sufu-cKO mice. This is further supported by NeuN immunostaining in the internal granule cell layer, which labels mature, non-diving neurons, and KI67 immunostaining, indicating dividing cells, and primarily found in the EGL. The mice were treated beginning at a timepoint when cerebellar cytoarchitecture was shown to be disrupted and it is indistinguishable from control following treatment. Fig.3 presents the most convincing and exciting data in this manuscript.

Sufu-cKO do not readily develop cerebellar tumors. The authors detected phosphorylated H2AX immunostaining, which labels double strand breaks, was in some cells in the EGL in regions of cerebellar dysgenesis in the Sufu-cKO, as was cleaved Caspase 3, a marker of apoptosis. P53, downstream of the double strand break pathway, protein was reduced in Sufu-cKO cerebellum. Genetically removing p53 from the Sufu-cKO cerebellum resulted in cerebellar tumors in 2 mo mice. The Sufu;p53-dKO cerebella at P0 lacked clear foliation, and the secondary fissure, even more so than the Sufu-cKO. Fgf5 RNA and signaling (pERK1/2) were also expressed ectopically.

In the revised manuscript, additional details have been added to clarify statistical analyses and to state limitations of the reported results in the absence of further experimental analyses.

---

## [Referee Report · Reviewer #2 (Public review)]

Summary:

Mutations in SUFU are implicated in SHH medulloblastoma (MB). SUFU modulates Shh signaling in a context-dependent manner, making its role in MB pathology complex and not fully understood. This study reports that elevated FGF5 levels are associated with a specific subtype of SHH MB, particularly in pediatric cases. The authors demonstrate that Sufu deletion in a mouse model leads to abnormal proliferation of granule cell precursors (GCPs) at the secondary fissure (region B), correlating with increased Fgf5 expression. Notably, pharmacological inhibition of FGFR restores normal cerebellar development in Sufu mutant mice.

Strengths:

The identification of increased FGF5 in subsets of MB is novel and a key strength of the paper.

Weaknesses:

The study appears incomplete despite the potential significance of these findings. The current paper does not fully establish the causal relationship between Fgf5 and abnormal cerebellar development, nor does it clarify its connection to SUFU-related MB. Some conclusions seem overstated, and the central question of whether FGFR inhibition can prevent tumor formation remains untested.

Comments on revisions:

In this revised version, many of the concerns and comments raised by this and other reviewers remain unaddressed and require attention in future studies. The manuscript does not demonstrate significant improvement.

Specific Comments:

(1) In the figure provided by the authors, FGF5 appears to be highly expressed beneath the GCPs. Regarding our comment and Reviewer 1's Comment 7, it is essential to identify the cell types secreting FGF5 and clarify whether it functions in a paracrine or autocrine manner. This should be incorporated into the working model illustrated in Figure 5.

(2) Contrary to the authors' claim that their results align completely with Jiwani et al. (2020), there is a discrepancy in the data. Jiwani et al. reported an increase in Gli2 levels in the Sufu mutant, whereas the current study shows the opposite result. This inconsistency needs to be addressed.

---

## [Referee Report · Reviewer #3 (Public review)]

Summary:

The interaction between FGF signaling and SHH-mediated GNP expansion in MB, particularly in the context of Sufu LoF, has just begun to be understood. The manuscript by Yabut et al. establishes a connection between ectopic FGF5 expression and GNP over-expansion in a late stage embryonic Sufu LoF model. The data provided links region-specific interaction between aberrant FGF5 signaling with SHH subtype of medulloblastoma. New data from Yabut et al. suggest that ectopic FGF5 expression correlates with GNP expansion near the secondary fissure in Sufu LoF cerebella. Furthermore, pharmacological blockade of FGF signaling inhibits GNP proliferation. Interestingly, the data indicate that the timing of conditional Sufu deletion (E13.5 using the hGFAP-Cre line) results in different outcomes compared to later deletion (using Math1-cre line, Jiwani et al., 2020). This study provides significant insights into the molecular mechanisms driving GNP expansion in SHH subgroup MB, particularly in the context of Sufu LoF. It highlights the potential of targeting FGF5 signaling as a therapeutic strategy. Additionally, the research offers a model for better understanding MB subtypes and developing targeted treatments.

Strengths:

One notable strength of this study is the extraction and analysis of ectopic FGF5 expression from a subset of MB patient tumor samples. This translational aspect of the study enhances its relevance to human disease. By correlating findings from mouse models with patient data, the authors strengthen the validity of their conclusions and highlight the potential clinical implications of targeting FGF5 in MB therapy.

The data convincingly show that FGFR signaling activation drives GNP proliferation in Sufu conditional knockout models. This finding is supported by robust experimental evidence, including pharmacological blockade of FGF signaling, which effectively inhibits GNP proliferation. The clear demonstration of a functional link between FGFR signaling and GNP expansion underscores the potential of FGFR as a therapeutic target in SHH subgroup medulloblastoma.

Previous studies have demonstrated the inhibitory effect of FGF2 on tumor cell proliferation in certain MB types, such as the ptc mutant (Fogarty et al., 2006)(Yaguchi et al., 2009). Findings in this manuscript provide additional support suggesting multiple roles for FGF signaling in cerebellar patterning and development.

Weaknesses:

In the GEO dataset analysis, where FGF5 expression is extracted, the reporting of the P-value lacks detail on the statistical methods used, such as whether an ANOVA or t-test was employed. Providing comprehensive statistical methodologies is crucial for assessing the rigor and reproducibility of the results. The absence of this information raises concerns about the robustness of the statistical analysis.

Another concern is related to the controls used in the study. Cre recombinase induces double-strand DNA breaks within the loxP sites, and the control mice did not carry the Cre transgene (as stated in the Method section), while Sufu-cKO mice did. This discrepancy necessitates an additional control group to evaluate the effects of Cre-induced double-strand breaks on phosphorylated H2AX-DSB signaling. Including this control would strengthen the validity of the findings by ensuring that observed effects are not artifacts of Cre recombinase activity.

Although the use of the hGFAP-Cre line allows genetic access to late embryonic stage, this also targets multiple cell types, including both GNPs and cerebellar glial cells. However, the authors focus primarily on GNPs without fully addressing the potential contributions of neuron-glial interaction. This oversight could limit the understanding of the broader cellular context in which FGF signaling influences tumor development.

- Statistical analysis from the Geo expression dataset:

The reviewer is satisfied with the revisions provided by the author and considers Figure 1 substantially improved.

- Generation of Sufu-cKO;Gli1-LacZ triple transgenic mice not described:

>The reviewer finds that the supplementary Figure 1 revisions provided by the author do not fully address the concerns raised, and the issue remains unresolved.

- Request control group to evaluate the effects of Cre induced double-strand breaks on phosphorylated H2AX-DSB signaling:

>Despite the revisions, control group (hGFAPCre;Sufu-fl/+) highlighted in the author response does not present in the revision, leaving this issue unresolved.

- hGFAP-Cre line also targets multiple celltypes, including both GNPs and cerebellar glial cells:

>The author acknowledges the limitations of the study, and the reviewer concurs, noting that it enhances the contextual understanding of the findings.

---

## [Author Response]

The following is the authors’ response to the original reviews.

The reviewers suggest a number of experiments and re-analyses to strengthen their claims and enhance the impact of the study. While a number of these are longer term, below is a summary of experiments and analyses recommended by the reviewers that can be accomplished in the shorter term:(1) Clarification of statistical approaches, quantification, data presentation and description of cerebellar anatomical nomenclature (e.gs. detailed statistical methods for the GEO dataset analysis, FDR correction, quantification in Figs 2-4)

The revised manuscript will provide detailed statistical methods including FDR correction for GEO dataset analyses and quantification. Please see specific responses to GEO dataset analyses below.

(2) Improved quality of images for select immunostains and in situ hybridization

The revised manuscript will address the quality of the images as indicated by the reviewers.

(3) Include a control group of hGFAP-Cre mice with loxP sites but without Sufu deletion to assess the effects of Cre-induced double-strand breaks on phosphorylated H2AX-DSB signaling.

The breeding scheme we used to generate homozygous SUFU conditional mutants will not generate pups carrying only hGFAP-Cre. Thus, we are unable to compare expression of gH2AX expression in littermates that do not carry loxP sites. The reviewer is correct in pointing out the possibility of Cre recombinase activity inducing double-strand breaks on its own. However, it is likely that any hGFAP-Cre induced double-strand breaks does not sufficiently cause the phenotypes we observed in homozygous mutants (Sufu-cKO) mice because the cerebellum of mice carry heterozygous SUFU mutations (hGFAP-Cre;Sufu-fl/+) do not display the profound cerebellar phenotypes observed in Sufu-cKO mice. We cannot rule out, however, any undetectable abnormalities that could be present which may require further analyses.

**Public Reviews:**

**Reviewer #1 (Public Review):**
Summary:SUFU modulates Sonic hedgehog (SHH) signaling and is frequently mutated in the B-subtype of SHH-driven medulloblastoma. The B-subtype occurs mostly in infants, is often metastatic, and lacks specific treatment. Yabut et al. found that Fgf5 was highly expressed in the B-subtype of SHH-driven medulloblastoma by examining a published microarray expression dataset. They then investigated how Fgf5 functions in the cerebellum of mice that have embryonic Sufu loss of function. This loss was induced using the hGFAP-cre transgene, which is expressed in multiple cell types in the developing cerebellum, including granule neuron precursors (GNPs) derived from the rhombic lip. By measuring the area of Pax6+ cells in the external granule cell layer (EGL) of Sufu-cKO mice at postnatal day 0, they find Pax6+ cells occupy a larger area in the posterior lobe adjacent to the secondary fissure, which is poorly defined. They show that Fgf5 RNA and phosphoErk1/2 immunostaining are also higher in the same disrupted region. Some of the phosphoErk1/2+ cells are proliferative in the Sufu-cKO. Western blot analysis of Gli proteins that modulate SHH signaling found reduced expression and absence of Gli1 activity in the region of cerebellar dysgenesis in Sufu-cKO mice. This suggests the GNP expansion in this region is independent of SHH signaling. Amazingly, intraventricular injection of the FGFR1-2 antagonist AZD4547 from P0-4 and examined histologically at P7 found the treatment restored cytoarchitecture in the cerebella of Sufu-cKO mice. This is further supported by NeuN immunostaining in the internal granule cell layer, which labels mature, non-diving neurons, and KI67 immunostaining, indicating dividing cells, and primarily found in the EGL. The mice were treated beginning at a timepoint when cerebellar cytoarchitecture was shown to be disrupted and it is indistinguishable from control following treatment. Figure 3 presents the most convincing and exciting data in this manuscript.Sufu-cKO do not readily develop cerebellar tumors. The authors detected phosphorylated H2AX immunostaining, which labels double-strand breaks, in some cells in the EGL in regions of cerebellar dysgenesis in the Sufu-cKO, as was cleaved Caspase 3, a marker of apoptosis. P53, downstream of the double-strand break pathway, the protein was reduced in Sufu-cKO cerebellum. Genetically removing p53 from the Sufu-cKO cerebellum resulted in cerebellar tumors in 2-month old mice. The Sufu;p53-dKO cerebella at P0 lacked clear foliation, and the secondary fissure, even more so than the Sufu-cKO. Fgf5 RNA and signaling (pERK1/2) were also expressed ectopically.The conclusions of the paper are largely supported by the data, but some data analysis need to be clarified and extended.(1) The rationale for examining Fgf5 in medulloblastoma is not sufficiently convincing. The authors previously reported that Fgf15 was upregulated in neocortical progenitors of mice with conditional loss of Sufu (PMID: 32737167). In Figure 1, the authors report FGF5 expression is higher in SHH-type medulloblastoma, especially the beta and gamma subtypes mostly found in infants. These data were derived from a genome-wide dataset and are shown without correction for multiple testing, including other Fgfs. Showing the expression of other Fgfs with FDR correction would better substantiate their choice or moving this figure to later in the manuscript as support for their mouse investigations would be more convincing.

To assess FGF5 (ENSG00000138675) expression in MB tissues, we used Geo2R (Barrett et al., 2013) to analyze published human MB subtype expression arrays from accession no. GSE85217 (Cavalli et al., 2017). GEO2R is an interactive web tool that compares expression levels of genes of interest (GOI) between sample groups in the GEO series using original submitter-supplied processed data tables. We entered the GOI Ensembl ID and organized data sets according to age and MB subgroup or MB^SHH^ subtype classifications. GEO2R results presented gene expression levels as a table ordered by FDR-adjusted (Benjamini & Hochberg) p-values, with significance level cut-off at 0.05, processed by GEO2R’s built-in limma statistical test. Resulting data were subsequently exported into Prism (GraphPad). We generated scatter plots presenting FGF5 expression levels across all MB subgroups (Figure 1A) and MB^SHH^ subtypes (Figure 1D). We performed additional statistical analyses to compare FGF5 expression levels between MB subgroups and MB^SHH^ subtypes and graphed these data as violin plots (Figure 1B, 1C, and 1E). For these analyses, we used one-way ANOVA with Holm-Sidak’s multiple comparisons test, single pooled variance. *P* value ≤0.05 was considered statistically significant. Graphs display the mean ± standard error of the mean (SEM).

**Author response image 1. sa4fig1:** Comparative expression of FGF ligands, FGF5, FGF10, FGF12, and FGF19, across all MB subgroups. FGF12 expression is not significantly different, while FGF5, FGF10, and FGF19, show distinct upregulation in MB^SHH subgroup (MBWNT n=70, MBSHH n=224, MBGR3 n=143, MBGR4 n=326).^

Expression of the 21 known FGF ligands were also analyzed. Many FGFs did not exhibit differential expression levels in MB^SHH^ compared to other MB subgroups, such as with FGF12 in Figure 1. FGF5, FGF10, and FGF19 (the human orthologue of mouse FGF15) all showed specific upregulation in MB^SHH^ compared to other MB subgroups (Author response image 1), supporting our previous observations that FGF15 is a downstream target of SHH signaling (Yabut et al., 2020), as the reviewer pointed out. However, further stratification of MB^SHH^ patient data revealed that only FGF5 specifically showed upregulation in infants with MBSHH (MB^SHHβ^ and MB^SHHγ^ Author response image 2) indicating a more prominent role for FGF5 in the developing cerebellum and driver of MB^SHH^ tumorigenesis in this dynamic environment.

**Author response image 2. sa4fig2:** Comparative expression of FGF5, FGF10, and FGF19 in different MB^SHH^ subtypes. FGF5 specifically show mRNA relative levels above 6 in 81% of MB^SHH^ infant patient tumors (n=80 MB^SHHα^ and MB^SHHγ^ tumors) unlike 35% of MB^SHHα^ (n=65) or 0% of MB^SHHδ^ (n=75) tumors.

(2) The Sufu-cKO cerebellum lacks a clear anchor point at the secondary fissure and foliation is disrupted in the central and posterior lobes. It would be helpful for the authors to review Sudarov & Joyner (PMID: 18053187) for nomenclature specific to the developing cerebellum.

The reviewers are correct that the cerebellar foliation is severely disrupted in central and posterior lobes, as per Sudarov and Joyner (Neural Development 2007). This nomenclature may be referred to describe the regions referred in this manuscript.

(3) The metrics used to quantify cerebellar perimeter and immunostaining are not sufficiently described. It is unclear whether the individual points in the bar graph represent a single section from independent mice, or multiple sections from the same mice. For example, in Figures 2B-D. This also applies to Figure 3C-D.

All quantification were performed from 2-3 20 um cerebellar sections of 3-6 independent mice per genotype analyzed. Individual points in the bar graphs represent the average cell number (quantified from 2-3 sections) from each mice. Figure 2B show data points from n=4 mice per genotype. Figure 2C show data from n=3 mice per genotype. Figure 2D show data from n=6 mice per genotype. Figure 3C-D show data from n=3 mice per genotype.

(4) The data on Fgf5 RNA expression presented in Figure 2E are not sufficiently convincing. The perimeter and cytoarchitecture of the cerebellum are difficult to see and the higher magnification shown in 2F should be indicated in 2E.

The lack of foliation in Sufu-cKO cerebellum is clear particularly when visualizing the perimeter via DAPI labeling (Figure 2E). The expression area of FGF5 is also visibly larger, given that all images in Figure 2E are presented in the same scale (scale bars = 500 um).

(5) The data presented in Figure 3 are not sufficiently convincing. The number of cells double positive for pErk and KI67 (Figure 3B) are difficult to see and appear to be few, suggesting the quantification may be unreliable.

We used KI67+ expression to provide a molecular marker of regions to be quantified in both WT and Sufu-cKO sections. Quantification of labeled cells were performed in images obtained by confocal microscopy, enabling imaging of 1-2 um optical slices since Ki67 or pERK expression might not localize within the same cellular compartments. We relied on continuous DAPI nuclear staining to distinguish individual cells in each optical slice and the colocalization of of Ki67 and pERK. All quantification were performed from 2-3 20 um cerebellar sections of 3-6 independent mice per genotype analyzed. Individual points in the bar graphs represent the average cell number (quantified from 2-3 sections) from each mice.

(6) The data presented in Figure 4F-J would be more convincing with quantification. The Sufu;p53-dKO appears to have a thickened EGL across the entire vermis perimeter, and very little foliation, relative to control and single cKO cerebella. This is a more widespread effect than the more localized foliation disruption in the Sufu-cKO.

We agree with the reviewers that quantification of these phenotypes provide a solid measure of the defects. The phenotypes of Sufu:p53-dKO cerebellum are so profound requiring in-depth characterization that will be the focus of future studies.

(7) Figure 5 does not convincingly summarize the results. Blue and purple cells in sagittal cartoon are not defined. Which cells express Fgf5 (or other Fgfs) has not been determined. The yellow cells are not defined in relation to the initial cartoon on the left.

The revised manuscript will address this confusion by clearly labeling the cells and their roles in the schematic diagram.

**Reviewer #2 (Public Review):**
Summary:Mutations in SUFU are implicated in SHH medulloblastoma (MB). SUFU modulates Shh signaling in a context-dependent manner, making its role in MB pathology complex and not fully understood. This study reports that elevated FGF5 levels are associated with a specific subtype of SHH MB, particularly in pediatric cases. The authors demonstrate that Sufu deletion in a mouse model leads to abnormal proliferation of granule cell precursors (GCPs) at the secondary fissure (region B), correlating with increased Fgf5 expression. Notably, pharmacological inhibition of FGFR restores normal cerebellar development in Sufu mutant mice.Strengths:The identification of increased FGF5 in subsets of MB is novel and a key strength of the paper.Weaknesses:The study appears incomplete despite the potential significance of these findings. The current paper does not fully establish the causal relationship between Fgf5 and abnormal cerebellar development, nor does it clarify its connection to SUFU-related MB. Some conclusions seem overstated, and the central question of whether FGFR inhibition can prevent tumor formation remains untested.
**Reviewer #3 (Public Review):**
Summary:The interaction between FGF signaling and SHH-mediated GNP expansion in MB, particularly in the context of Sufu LoF, has just begun to be understood. The manuscript by Yabut et al. establishes a connection between ectopic FGF5 expression and GNP over-expansion in a late-stage embryonic Sufu LoF model. The data provided links region-specific interaction between aberrant FGF5 signaling with the SHH subtype of medulloblastoma. New data from Yabut et al. suggest that ectopic FGF5 expression correlates with GNP expansion near the secondary fissure in Sufu LoF cerebella. Furthermore, pharmacological blockade of FGF signaling inhibits GNP proliferation. Interestingly, the data indicate that the timing of conditional Sufu deletion (E13.5 using the hGFAP-Cre line) results in different outcomes compared to later deletion (using Math1-cre line, Jiwani et al., 2020). This study provides significant insights into the molecular mechanisms driving GNP expansion in SHH subgroup MB, particularly in the context of Sufu LoF. It highlights the potential of targeting FGF5 signaling as a therapeutic strategy. Additionally, the research offers a model for better understanding MB subtypes and developing targeted treatments.Strengths:One notable strength of this study is the extraction and analysis of ectopic FGF5 expression from a subset of MB patient tumor samples. This translational aspect of the study enhances its relevance to human disease. By correlating findings from mouse models with patient data, the authors strengthen the validity of their conclusions and highlight the potential clinical implications of targeting FGF5 in MB therapy.The data convincingly show that FGFR signaling activation drives GNP proliferation in Sufu, conditional knockout models. This finding is supported by robust experimental evidence, including pharmacological blockade of FGF signaling, which effectively inhibits GNP proliferation. The clear demonstration of a functional link between FGFR signaling and GNP expansion underscores the potential of FGFR as a therapeutic target in SHH subgroup medulloblastoma.Previous studies have demonstrated the inhibitory effect of FGF2 on tumor cell proliferation in certain MB types, such as the ptc mutant (Fogarty et al., 2006)(Yaguchi et al., 2009). Findings in this manuscript provide additional support suggesting multiple roles for FGF signaling in cerebellar patterning and development.Weaknesses:In the GEO dataset analysis, where FGF5 expression is extracted, the reporting of the P-value lacks detail on the statistical methods used, such as whether an ANOVA or t-test was employed. Providing comprehensive statistical methodologies is crucial for assessing the rigor and reproducibility of the results. The absence of this information raises concerns about the robustness of the statistical analysis.

The revised manuscript will include the following detailed explanation of the statistical analyses of the GEO dataset:

For the analysis of expression values of FGF5 (ENSG00000138675), we obtained these values using Geo2R (Barrett et al., 2013), which directly analyze published human MB subtype expression arrays from accession no. GSE85217 (Cavalli et al., 2017). GEO2R is an interactive web tool that compares expression levels of genes of interest (GOI) between sample groups in the GEO series using original submitter-supplied processed data tables. We simply entered the GOI Ensembl ID and organized data sets according to age and MB subgroup or MBSHH subtype classifications. GEO2R results presented gene expression levels as a table ordered by FDR-adjusted (Benjamini & Hochberg) p-values, with significance level cut-off at 0.05, processed by GEO2R’s built-in limma statistical test. Resulting data were subsequently exported into Prism (GraphPad). We generated scatter plots presenting FGF5 expression levels across all MB subgroups (Figure 1A) and MBSHH subtypes (Figure 1D). We performed additional statistical analyses to compare FGF5 expression levels between MB subgroups and MBSHH subtypes and graphed these data as violin plots (Figure 1B, 1C, and 1E). For these analyses, we used one-way ANOVA with Holm-Sidak’s multiple comparisons test, single pooled variance. *P* value ≤0.05 was considered statistically significant. Graphs display the mean ± standard error of the mean (SEM). Sample sizes wereAuthor response table 1:

**Author response table 1. sa4table1:** 

MB^WNT^ n=70	MB^SHHα^ n=66
MB^SHH^ n=224	MB^SHHβ^ n=35
MB^GR3^ n=143	MB^SHHγ^ n=47
MB^GR4^ n=326	MB^SHHδ^ n=77

Another concern is related to the controls used in the study. Cre recombinase induces double-strand DNA breaks within the loxP sites, and the control mice did not carry the Cre transgene (as stated in the Method section), while Sufu-cKO mice did. This discrepancy necessitates an additional control group to evaluate the effects of Cre-induced double-strand breaks on phosphorylated H2AX-DSB signaling. Including this control would strengthen the validity of the findings by ensuring that observed effects are not artifacts of Cre recombinase activity.

The breeding scheme we used to generate homozygous SUFU conditional mutants will not generate pups carrying only hGFAP-Cre. Thus, we are unable to compare expression of gH2AX expression in littermates that do not carry loxP sites. The reviewer is correct in pointing out the possibility of Cre recombinase activity inducing double-strand breaks on its own. However, it is likely that any hGFAP-Cre induced double-strand breaks does not sufficiently cause the phenotypes we observed in homozygous mutants (Sufu-cKO) mice because the cerebellum of mice carry heterozygous SUFU mutations (hGFAP-Cre;Sufu-fl/+) do not display the profound cerebellar phenotypes observed in Sufu-cKO mice. We cannot rule out, however, any undetectable abnormalities that could be present which may require further analyses.

Although the use of the hGFAP-Cre line allows genetic access to the late embryonic stage, this also targets multiple celltypes, including both GNPs and cerebellar glial cells. However, the authors focus primarily on GNPs without fully addressing the potential contributions of neuron-glial interaction. This oversight could limit the understanding of the broader cellular context in which FGF signaling influences tumor development.

The reviewer is correct in that hGFAP-Cre also targets other cell types, such as cerebellar glial cells, which are generated when Cre-expression has begun. It is possible that cerebellar glial cell development is also compromised in Sufu-cKO mice and may disrupt neuron-glial interaction, due to or independently of FGF signaling. In-depth studies are required to interrogate how loss of SUFU specifically affect development of cerebellar glial cells and influence their cellular interactions in the developing cerebellum.

**Recommendations for the authors:**

**Editorial Comments:**
The reviewers suggest a number of steps to improve the manuscript that include additional experiments and a deeper analyses and re-evaluation of existing data. Short of significant new experiments, there appears to be number of straightforward analyses that can improve the study:(1) Reanalyses of statistical and quantitative approaches used e.gs FDR correction, cerebellar deficits, GEO analyses.

The revised manuscript will include detailed information on the statistical and quantitative approaches as addressed in our response to the reviewer’s comments.

(2) More clear presentation of qualitative labeling approaches (immunohistochemistry and in situ hybridization).

A detailed description of the protocols used will be included in the methods section for labeling methods in the revised manuscript.

**Reviewer #1 (Recommendations For The Authors):**
AZD4547 treatment of the dKO mice would provide more convincing evidence that FGF-targeted treatments could curtail tumor growth in these mice or refute the suggestion that FGF-targeted treatment could prevent tumor growth.

We agree that performing AZD4547 treatment on Sufu-dKO mice will strengthen these studies. However, we are unable to address since these mice are now unavailable. We hope that future studies will address these.

Atoh1 is referred to as Math1 (older nomenclature) and should be corrected.

The revised manuscript will include this change in nomenclature.

Check verb tense throughout the manuscript.

We will edit the manuscript further to check verb tenses prior to submission of the revised manuscript.

**Reviewer #2 (Recommendations For The Authors):**
Specific Comments:(1) The identification of increased FGF5 in subsets of MB is novel and a key strength of the paper. However, the causal relationship between FGF5 and MB remains unestablished. Based on the current data, FGF5 can only be considered a biomarker for stratifying MB.

We agree with the reviewer that our studies do not provide direct evidence that FGF5 cause MB. Future investigation focusing on determining if FGF5 inhibition leads to phenotypic rescue will strongly establish the relationship between FGF5 and MB. The reviewer is also correct that our studies reveal that FGF5 acts as a potential biomarker, as we mentioned in the Discussion section.

(2) The upregulation of Fgf5 in Sufu-deficient cerebella is crucial to this study, yet the presented data are unconvincing to support this conclusion. In comparing Fgf5 expression between WT and Sufu mutants (Figures 2E, F and 4I), the cerebellar sections differ significantly, with mutant sections seemingly from a more lateral position. The authors should provide images of mutant sections from more comparable positions to accurately assess the effect of Sufu deficiency on Fgf5 expression. Additionally, the signals in Figure 2F resemble non-specific backgrounds rather than specific RNAscope signals.

The WT and mutant sections analyzed were carefully selected from comparable levels. The abnormal foliation in Sufu-cKO make the mutant sections look like they are from the lateral cerebellum.

Figure 2F (enlarged regions) point to punctate RNAScope signals which is characteristic of this labeling method (see RBFOX3 or GFAP labeling in DAPI-labeled cells in the mouse brain at https://acdbio.com/science/applications/research-areas/neuroscience). The higher number of punctate signals in some, but not all, DAPI-labeled cells in Figure 2F indicate that the FGF5 RNAScope signal is specific.

(3) Jiwani et al. (2020) reported that Fgf8 also expressed in region B of the EGL, is upregulated in Sufu-deficient cerebella and is necessary and sufficient for Sufu mutant GCP proliferation. The current study does not distinguish whether the FGFR inhibitor AZD4547 blocks Fgf5 and Fgf8 function in restoring cerebellar histology in Sufu mutants.

AZD4547 potently inhibits FGFR1, FGFR2, and FGFR3 autophosphorylation (Gavine et al., Cancer Research, 2012). FGF8 is reported to bind to these receptors (Ornitz and Itoh, 2015). Thus, the reviewer is correct that the studies will not distinguish between FGF5 or FGF8 activity. Further investigation on FGF8 expression and the effects of its inhibition in the Sufu-cKO neonatal cerebellum will determine whether tumorigenic processes are driven by either FGF5 or FGF8. Nevertheless, we postulate that FGF5 is exerting a greater effect in activating FGF signaling in the developing cerebellum given that it is highly expressed along the external granule layers of the developing cerebellum (Author response image 3).

**Author response image 3. sa4fig3:** Expression of FGF5 and FGF8 in the P4 mouse cerebellum (Allen Brain Atlas, https://developingmouse.brain-map.org).

(4) The authors should show whether AZD4547 treatment restores normal Fgf5 expression. Importantly, they need to test whether AZD4547 rescues the proliferation defect observed in Sufu;p53 double mutants.

We agree that performing AZD4547 treatment on Sufu-dKO mice will strengthen these studies. However, we are unable to address since these mice are now unavailable. We hope that future studies will address these.

(5) Jiwani et al. (2020) showed that deleting Sufu with Atoh1-Cre promotes Gli3R and suppresses Gli2 levels, leading to increased cell proliferation and delayed cell cycle exit in the central lobe. The findings of the current study (Supplementary Figure 1) seem to differ from this previous report, yet both studies conclude that Sufu-KO disrupts differentiation. The authors should provide an explanation for this discrepancy.

Our results align completely with the findings by Jiwani et al. (2020). Both studies showed reduced levels of Gli3R, showing nearly 50% reduction, when Sufu is deleted (see Figure 4A-4D in Jiwani et al., 2020).

(6) The hGFAP-Cre mouse line is used to delete Sufu from the cerebellum, but it is not commonly used for GCP-specific deletion. The authors need to provide a reference or more details on the temporal and spatial activity of the Cre line, as the cited paper describes its generation but offers little information on its activity in the developing cerebellum.

We appreciate the reviewer’s reminder to include the reference for the Schuller et al. 2008 paper. This study characterized the hGFAP-Cre temporal and spatial expression in the developing cerebellum, including granule cell precursors. We will include this reference in the revised manuscript.

(7) Based on the provided data, it is difficult to determine which cell types express Fgf5. Given that hGFAP-cre may delete Sufu in other cerebellar cell types, the authors should demonstrate that Fgf5 is expressed in granule cells or granule cell precursors.

Future studies will focus on further characterization of the role of FGF5 in cerebellar development, including the identity cells expressing FGF5. The reviewer is correct in that hGFAP-Cre also targets other cell types and that Sufu deletion in these cells induced ectopic FGF5 expression.

(8) The provided data show an increase in pERK+ cells in GCPs at the secondary fissure. This increase may simply reflect an accumulation of GCPs. It is unconvincing that there is an increase in pERK due to the loss of Sufu.

The reviewer is correct that the increase in GCPs will also result increase the number of pERK+ cells. To control for this, our quantification reflects the number of cells per unit area where Ki67+ cells. With these parameters, we found that there is an increased density of pERK+ cells in a given Ki67+ region. All quantification were performed from 2-3 20 um cerebellar sections of 3-6 independent mice per genotype analyzed. Individual points in the bar graphs represent the average cell number (quantified from 2-3 sections) from each mice.

(9) No data are provided on MB formation in Sufu-cKO; p53- mutants, and it is unknown whether FGFR inhibitors block tumor formation.

We agree that performing AZD4547 treatment on Sufu-dKO mice will strengthen these studies. However, we are unable to address since these mice are now unavailable. We hope that future studies will address these.

(10) The authors frequently mention "preneoplastic lesions" of GCPs in Sufu mutant mice. What evidence supports this claim?

Preneoplastic lesions are defined as cells carrying genetic and phenotypic alterations that show higher risk of malignancy (such as MB) but lack the capacity to grow autonomously in the absence of a secondary factor (Feo et al., 2011). In Sufu-cKO mice, we see abnormally proliferating and behaving granule precursor cells that do not grow autonomously, in the absence of a p53 LOF. The combined deletion of Sufu and p53 transforms these cells to become neoplastic.

(11) Fgf5 is normally expressed in region B. What is its potential function? Does AZD4547 affect normal development?

Future studies will focus on further characterization of the role of FGF5 in cerebellar development, including the identity cells expressing FGF5. Regarding AZD4547, we did not observe any obvious difference between AZD4547-treated and vehicle-treated cerebelli. These indicate that AZD4547 inhibition of FGFRs under physiologic conditions does not significantly disrupt normal cerebellar development.

(12) Figure 3G: It is unclear which specimens were treated with AZD4547. The authors mention treatment in line 281 but contradict themselves in the figure legend.

We thank the reviewer for pointing out this typo. Cerebellar tissues shown in Figure 3G were all treated with AZD4547. The figure legend will be corrected in the revised manuscript.

(13) Figure 4J: The higher magnification images of the pERK/Ki67 staining appear identical in the control and Sufu;p53-dKO. The authors need to correct the mistake.

We thank the reviewer for pointing this out. We will correct this figure in the revised manuscript.

Minor Comments:(1) Whenever possible, images comparing WT and mutants should be presented at the same scale within a figure. For example, readers might easily conclude that mutant brains are smaller than controls in Figure 4G.

Unfortunately, because the cerebellum of Sufu;p53-dKO mice are significantly bigger, we are unable to show the whole cerebellum in the same scale in Figure 4G. We wanted to emphasize the significant and abnormal cerebellar growth in this figure.

(2) The figure legend for Supplementary Figure 2 is missing.

Thank you for pointing this out. We will add a figure legend in this Supplementary data in the revised manuscript.

(3) The authors state that the expansion of Pax6+ GNPs in the newborn Sufu-cKO cerebellum (Figure 2) occurs in similar anatomical subregions where infantile MB tumors typically arise (Tan et al., 2018). The cited paper describes more abundant SHH MB in the cerebellar hemisphere. The authors need to elaborate on their statement to clarify this point.

The reviewer is correct in that Tan et al., 2018 observed tumors arising from the cerebellar hemisphere. More specifically, these tumors arise in the posterior/ventral regions of the cerebellar hemispheres (Figure 2 in Tan et al., 2018). Similarly, Sufu-cKO mice have more severe defects in the posterior/ventral regions of the cerebellar hemisphere (Figures 2A and 3F) and therefore corroborate the findings by Tan et al., that abnormal SHH signaling in these regions results in increased sensitivity to MB formation.

**Reviewer #3 (Recommendations For The Authors):**
Figure1 [Upregulated FGF5 expression in MBS-HH tumors]- Statistical analysis from the Geo expression dataset does not provide enough detail. At least, the authors should mention whether they have made any adjustments from the default settings and how they extracted/plotted the FGF5 expression (Figure 1BCE).

For the analysis of expression values of FGF5 (ENSG00000138675), we obtained these values using Geo2R (Barrett et al., 2013), which directly analyze published human MB subtype expression arrays from accession no. GSE85217 (Cavalli et al., 2017). GEO2R is an interactive web tool that compares expression levels of genes of interest (GOI) between sample groups in the GEO series using original submitter-supplied processed data tables. We simply entered the GOI Ensembl ID and organized data sets according to age and MB subgroup or MBSHH subtype classifications. GEO2R results presented gene expression levels as a table ordered by FDR-adjusted (Benjamini & Hochberg) p-values, with significance level cut-off at 0.05, processed by GEO2R’s built-in limma statistical test. Resulting data were subsequently exported into Prism (GraphPad). We generated scatter plots presenting FGF5 expression levels across all MB subgroups (Figure 1A) and MB^SHH^ subtypes (Figure 1D). We performed additional statistical analyses to compare FGF5 expression levels between MB subgroups and MB^SHH^ subtypes and graphed these data as violin plots (Figure 1B, 1C, and 1E). For these analyses, we used one-way ANOVA with Holm-Sidak’s multiple comparisons test, single pooled variance. *P* value ≤0.05 was considered statistically significant. Graphs display the mean ± standard error of the mean (SEM). See Author response table 1 for sample sizes.

Figure 3 [Ectopic activation of FGF signaling in the EGL of P0 Sufu-cKO cerebellum]- Gil1-lz mice reference wrong. Correct Bai CB, et al. 2002- Generation of Sufu-cKO;Gli1-LacZ triple transgenic mice not described- Veh vs. treated not labelled (Figure 3F)

We will address these minor text changes in the revised manuscript. A more detailed description of the generation of Sufu-cKO;Gli1-LacZ triple transgenic will also be included in the Methods section.

Figure 5 [Proposed model]- In the text, Figure 5 is mistaken for Figure 8.

We will address these minor text changes in the revised manuscript.